materials science

CoO nanoparticle, photocatalytic hydrogen production, g-C₃N₄

**Author for correspondence:**
Xuecheng Liu
e-mail: liuxc@ctbu.edu.cn

This article has been edited by the Royal Society of Chemistry, including the commissioning, peer review process and editorial aspects up to the point of acceptance.

# *In situ* growing of CoO nanoparticles on g-C$_3$N$_4$ composites with highly improved photocatalytic activity for hydrogen evolution

Xuecheng Liu[1,2], Qian Zhang[1], Liwei Liang[1], Lintao Chen[2], Yuyou Wang[1], Xiaoqing Tan[1], Li Wen[1] and Hongyu Huang[2]

[1]Engineering Research Center for Waste Oil Recovery Technology and Equipment, Chongqing Key Laboratory of Catalysis and New Environmental Materials, College of Environment and Resources, Chongqing Technology and Business University, Chongqing 400067, People's Republic of China
[2]Guangdong Provincial Key Laboratory of New and Renewable Energy Research and Development, Guangzhou Institute of Energy Conversion, Chinese Academy of Sciences, Guangzhou 510640, People's Republic of China

XL, 0000-0002-2260-8995

CoO/g-C$_3$N$_4$ hybrid catalyst is facilely prepared for application to photocatalytic H$_2$ evolution from water splitting by the vacuum rotation–evaporation and *in situ* thermal method. The physical and chemical properties of CoO/g-C$_3$N$_4$ are determined by a series of characterization methods. The g-C$_3$N$_4$ with 0.6 wt% Co loading exhibits superior photocatalytic hydrogen evolution activity with an H$_2$ evolution amount of 23.25 mmol g$^{-1}$ after 5 h. The obtained 0.6 wt% CoO/g-C$_3$N$_4$ can split water to generate 0.39 mmol g$^{-1}$ H$_2$ without sacrificial agent and noble metal, while the pure g-C$_3$N$_4$ is inactive under the same reaction conditions. The remarkable enhancement of photocatalytic H$_2$ evolution activity of CoO/g-C$_3$N$_4$ composites is mainly ascribed to the effective separation of electron–hole pairs and charge transfer. The work creates new opportunities for the design of low-cost g-C$_3$N$_4$-based photocatalysts with high photocatalytic H$_2$ evolution activity from overall water splitting.

# 1. Introduction

The development of photocatalytic water splitting into hydrogen by using solar energy has been considered as one of the most promising strategies in photocatalysis to solve the global energy crisis and environmental deterioration [1–3]. Great efforts have been devoted to searching for cheap and efficient photocatalysts. Graphitic carbon nitride (g-$C_3N_4$) with an appropriate band gap (approx. 2.7 eV) has been assumed to be a promising candidate photocatalyst for hydrogen production [4]. However, the photocatalytic hydrogen production activity of g-$C_3N_4$ is limited by low electrical conductivity, a lack of visible light absorption and high charge carries recombination rate [5,6].

The photocatalytic activity of materials can be improved with the enhanced band structures, light adsorbance, charge transport and photogenerated electron–hole pairs separation [7,8]. Several strategies, such as synthesis techniques [9], nanostructure design [10] and electronic structure modulation [11,12], have been conducted to obtain highly efficient g-$C_3N_4$-based photocatalysts. Apart from the above-mentioned methods, a Z-scheme photocatalytic system with two different photocatalysts which sparked interest from the natural photosynthetic systems of plant leaves is one of the most promising approaches to obtain hydrogen evolution from water splitting. The construction of Z-scheme g-$C_3N_4$-based composite has been widely investigated, e.g. NiO/g-$C_3N_4$ [13], $Cu_2O$/g-$C_3N_4$ [14], $CeO_2$/g-$C_3N_4$ [15], $TiO_2$/g-$C_3N_4$ [12], $Bi_2MoO_6$/g-$C_3N_4$ [16] and $Cr_2O_3$/g-$C_3N_4$ [17], to promote the photoactivity for water splitting. In comparison with pure g-$C_3N_4$, these Z-scheme g-$C_3N_4$-based composites have superior potential for promoting charge transportation, limiting the photogenerated electron–hole pairs recombination, strengthening light absorbance and lowering the redox overpotential [18]. Recently, the construction of heterojunction photocatalysts is mainly focused on how to effectively limit photogenerated electron–hole pairs recombination, and it places less emphasis on the selection of semiconductors. In fact, in order to optimize the fabrication of Z-scheme g-$C_3N_4$-based photocatalysts for overall water splitting, it is important to design a heterojunction photocatalyst with band energy alignments not only trapping an electron to effectively separate the photogenerated charges but also suppressing the back reaction of water formation.

Cobalt monoxide (CoO), as a traditional transition metal monoxide, has gained more attention for its application to photocatalytic water splitting. It is reported that CoO exhibits good photocatalytic water splitting activity with an extraordinary STH of around 5% [1,19]. Wang and co-workers [20] reported photocatalytic decomposition of water for hydrogen evolution on a CoO/$SrTiO_3$ catalyst in 2007. Besides, CoO with efficient photo-induced electrons separation can be used as an effective co-catalyst to improve the photocatalytic water splitting activity for hydrogen evolution [21]. But the poor stability of the CoO catalyst is caused by $H_2O_2$ poisoning, hindering its further development [22–24]. It is still a challenge to seek a suitable structure of CoO-based catalyst with high activity and stability.

It is reported that the combination of g-$C_3N_4$ and CoO can result in improved photocatalysts for water splitting [25–27]. The particles could be well dispersed on the carrier by the vacuum rotation–evaporation method [28]. In this work, CoO nanoparticles are growing *in situ* on the g-$C_3N_4$ to prepare well-dispersed CoO/g-$C_3N_4$ composite photocatalyst by the vacuum rotation–evaporation and thermal annealing methods under nitrogen atmosphere. The physical structure, chemical composition, photoelectric properties and photocatalytic $H_2$ generation activity of CoO/g-$C_3N_4$ nanocomposite with different Co loading are investigated in detail by a series of characterizations. The enhancement mechanism of photocatalytic overall water splitting for hydrogen evolution of as-synthesized CoO/g-$C_3N_4$ nanocomposite is also discussed.

# 2. Experimental section

## 2.1. Sample preparation

Urea, triethanolamine and cobalt nitrate with analytical grade are purchased from Aladdin Industrial Corporation (Shanghai, China). Firstly, g-$C_3N_4$ is prepared by the thermal polycondensation of urea [29]. Typically, 10 g urea is placed into a covered crucible and heated at 500°C for 3 h using a heating rate of 10°C min$^{-1}$ in a muffle furnace to obtain g-$C_3N_4$. By sonication, 200 mg g-$C_3N_4$ powder is dispersed in 50 ml of deionized water. According to the mass ratio of Co from 0 to 5%, the certain volume of Co($NO_3$)$_2$ aqueous solution is dipped into the prepared g-$C_3N_4$ aqueous dispersion and stirred continuously for 20 h to form homogeneous solution with water bath at 70°C for 12 h. After

rotavaporation to dryness, the obtained impregnated sample is annealed at 400°C for 2 h in nitrogen atmosphere in the tube furnace and the CoO nanoparticles are grown *in situ* on the g-$C_3N_4$ sheets to obtain CoO/g-$C_3N_4$ composites.

## 2.2. Sample characterization

The phase compositions of the prepared materials are determined by an X-ray diffractometer (XRD) with Cu Kα radiation (modelD/max RA, RigakuCo., Japan). The transmission electron microscope (TEM) images are obtained by using the electron microscope (JEM-6700F, Japan). X-ray photoelectron spectroscopy (XPS) measurements are analysed by Thermo ESCALAB 250, USA, with Al Ka X-rays radiation operated at 150 W. The XPS spectra of the samples were calibrated by using the C1s level at 284.5 eV as an internal standard. Diffuse reflectance spectra of the dry-pressed disc samples are performed by a UV–Visible spectrometer (UV-2700, Shimadzu, Japan). Photoluminescence (PL) is recorded on a fluorescence spectrometer with an Xe lamp as an excitation source with optical filters (FS-2500, Japan). Electrochemical analysis is carried out on a CHI660E workstation. Electrochemistry impedance spectroscopy (EIS) and photoelectric current response measurements are conducted on a conventional three-electrode system with the as-prepared photocatalyst coated onto a 2 cm × 1 cm FTO glass electrode as a working electrode, platinum foil as a counter electrode and Ag/AgCl as a reference electrode, respectively. The frequency range is from 0.01 Hz to 100 kHz in an alternating voltage of 0.02 V under chopped illumination with 40 s light on/off cycles in 0.1 M $Na_2SO_4$ aqueous solution. Incident light was performed by an Xe arc lamp.

## 2.3. Photocatalytic $H_2$ generation testing

Photocatalytic hydrogen evolution reactions are measured in a top-irradiation reaction vessel with a 300 W xenon lamp connected to a closed glass gas-circulation system (CEL-SPH2N, AG, CEAULIGHT). Fifty milligrams of photocatalyst are put into an aqueous solution with 45 ml water and 5 ml triethanolamine. Then, 1.5 wt% of Pt nanoparticles are loaded onto the surface of catalysts *in situ* photo-deposition by using $H_2PtCl_6\cdot6H_2O$ as the precursor. For the overall water splitting, 50 mg of photocatalyst is put into an aqueous solution with 50 ml water without sacrificial agent and noble metal Pt. Next, the reaction system is sealed and evacuated for 30 min to remove air prior to the irradiation experiments, and during the photocatalytic reaction, the reaction solution temperature is kept around 10°C by a flow of cooling water. The outlet gases are analysed by gas chromatography (GC 7920, Beijing) with a thermal conductivity detector and nitrogen as the carrier gas to determine the amount of generated hydrogen.

# 3. Result and discussion

The crystalline structures and the phase components of as-prepared CoO/g-$C_3N_4$ composites and g-$C_3N_4$ are studied by XRD. As shown in figure 1, the based materials give two typical diffraction peaks at 13.0° and 27.4°, which can be indexed to the (100) and (002) reflections of g-$C_3N_4$, respectively. It is assumed that g-$C_3N_4$ has a wrinkled sheet-like structure with relatively smooth surface [30]. For CoO/g-$C_3N_4$ composites, the diffraction peaks of g-$C_3N_4$ are observed clearly, indicating that these prepared samples maintain the basic structure of g-$C_3N_4$. But in comparison with pure g-$C_3N_4$, there is a distinct diffraction peak at 36.4°, which can perfectly match with the face-centred cubic CoO structure (JCPDS 71-1178). The characteristic diffraction peaks of both CoO and g-$C_3N_4$ reveal the successful fabrication of CoO/g-$C_3N_4$ composites by *in situ* growing of CoO nanoparticles on g-$C_3N_4$.

The TEM images of the prepared materials are presented in figure 2. As shown in figure 2b,c, the CoO nanoparticles are highly dispersed by *in situ* growing onto the g-$C_3N_4$ matrix. From the enlarged high-resolution TEM of 5% CoO/g-$C_3N_4$ in figure 2c, the exposed crystal planes of the obtained CoO can be easily observed, and the lattice fringes with a spacing of 0.25 nm are attributed to the (111) planes of CoO. Based on the XRD and TEM characterizations, this can provide solid evidence for the successful formation of a CoO/g-$C_3N_4$ heterostructure with the *in situ* growing method.

Surface chemical states of elements and the specific bonding in the prepared samples are characterized in-depth by XPS, and the results are shown in figure 3a–e. The full survey spectrum of the prepared material is shown in figure 3a. There are three sharp peaks with binding energy values

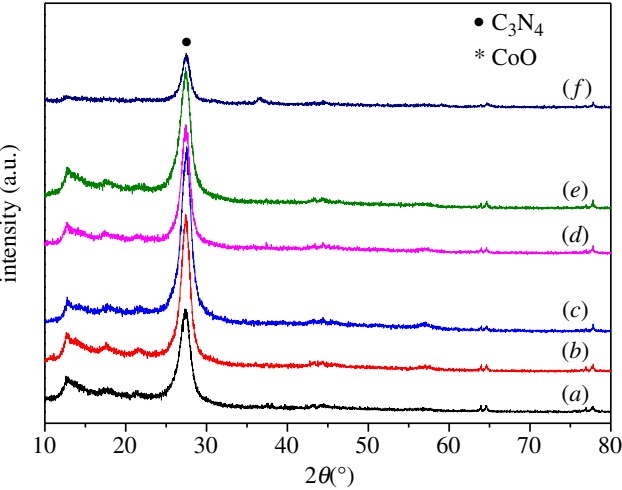

**Figure 1.** XRD patterns of (a) g-C₃N₄, (b) 0.3% CoO/g-C₃N₄, (c) 0.6% CoO/g-C₃N₄, (d) 0.8% CoO/g-C₃N₄, (e) 1% CoO/g-C₃N₄ and (f) 5% CoO/g-C₃N₄ composites.

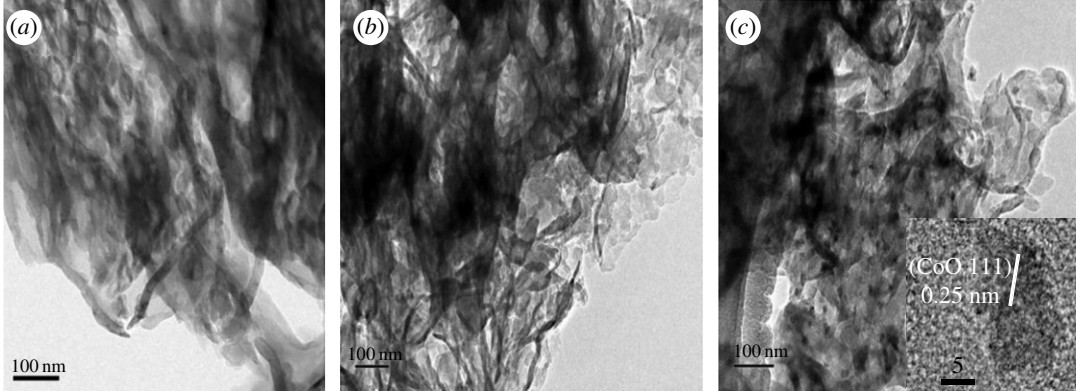

**Figure 2.** TEM and HR-TEM images of (a) g-C₃N₄, (b) 0.6% CoO/g-C₃N₄ and (c) 5% CoO/g-C₃N₄ composites.

of 285, 399, 530 and 782 eV attributed to the signals of C 1s, N 1s, O 1s and Co 2p, respectively, in the as-prepared samples. The C 1s spectra (figure 3b) can be deconvoluted into three peaks at 284.8 eV, 288.1 eV and 293.6 eV, respectively. The binding energy at 284.8 eV can be ascribed to the signals of C–C coordination of the surface adventitious carbon. The binding energy at 288.1 eV is attributed to the $sp^2$-hybridized carbon in N=C–N coordination, while the peak observed at 293.6 eV results from π-excitation [31]. Figure 3c presents the N 1s spectra, which can be subdivided into four peaks at 398.7, 399.4, 400.9 and 404.7 eV. The binding energy at 398.7 eV is ascribed to the $sp^2$-hybridized nitrogen atoms in C=N–C groups [32]. The binding energy at 399.4 eV is corresponding to the tertiary nitrogen N–C₃ groups or H–N–C₂ [33]. The binding energy at 400.9 eV is result from the amino function groups [32], and the binding energy at 404.7 eV is attributed to charging effects or positive charge localization in heterocycles [34]. The high-resolution XPS spectra of Co 2p of 0.6% CoO/g-C₃N₄ and 1% CoO/g-C₃N₄ are displayed in figure 3e. The weak and diffused Co 2p peaks of 0.6% CoO/g-C₃N₄ at two pairs of individual peaks centred at 780.3 and 796.2 eV confirm the existence of Co, which are identified as the major binding energies of Co²⁺ in CoO [35]. Two peaks at 780.6 and 796.5 eV can be attributed to the Co 2p₃/₂ and Co 2p₁/₂ spin-orbit peaks of CoO, respectively [1]. The O 1s spectra with two peaks at about 529 and 532 eV are shown in figure 3d. The binding energy at 529 eV is ascribed to the Co–O bond in the CoO phase [36], while the strong peak at about 532 eV corresponds to the Co–O–C bond, indicating that a strong interaction exists between CoO and g-C₃N₄ [26]. It can be seen that the signal of Co–O–C bond becomes bigger with the increase in Co loading because of the change of electronic state of adsorbed oxygen species [37]. Therefore, the interfacial interaction between CoO and g-C₃N₄ could be greatly enhanced due to the interfacial hybridized Co–O–C bond.

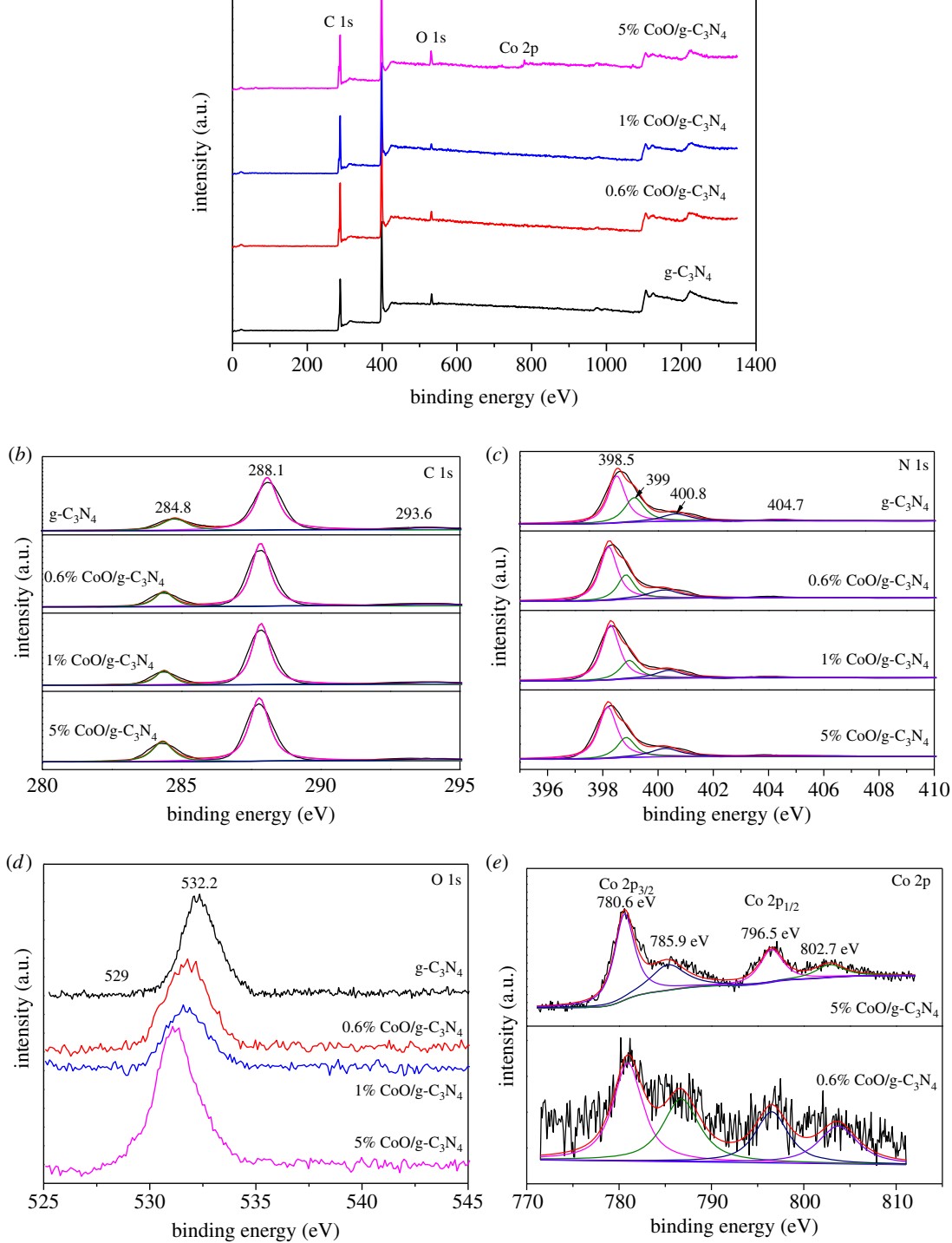

**Figure 3.** XPS profiles of (*a*) survey, (*b*) C 1s, (*c*) N 1s, (*d*) O 1s and (*e*) Co 2p of the prepared samples.

It is widely accepted that the optical and photoelectric properties are of great significance to the photocatalytic activity. UV–Vis absorption spectra and PL are measured to identify these properties of g-C$_3$N$_4$ and CoO/g-C$_3$N$_4$ composite samples. Figure 4 displays the UV–Vis diffuse reflectance spectra of pure g-C$_3$N$_4$ and CoO/g-C$_3$N$_4$ with different CoO contents. It is seen that 0.6% CoO/g-C$_3$N$_4$ exhibits the best ultraviolet and visible light absorbance, indicating that the 0.6% CoO/g-C$_3$N$_4$ composite could obtain the best photocatalytic activity for hydrogen evolution by using more solar light. The efficient separation of photo-induced electron–hole pairs is another factor to influence the photocatalytic activity. It is well known that photocatalytic activity is enhanced by interfaces of

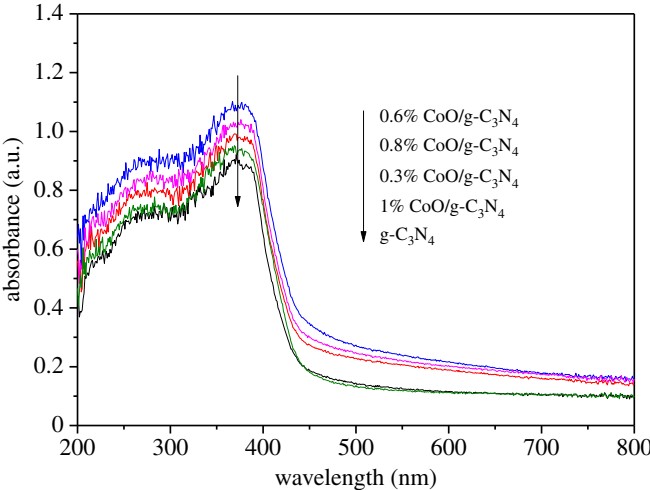

**Figure 4.** UV–Vis absorbance spectra of g-C$_3$N$_4$, 0.3% CoO/g-C$_3$N$_4$, 0.6% CoO/g-C$_3$N$_4$, 0.8% CoO/g-C$_3$N$_4$ and 1% CoO/g-C$_3$N$_4$ composites.

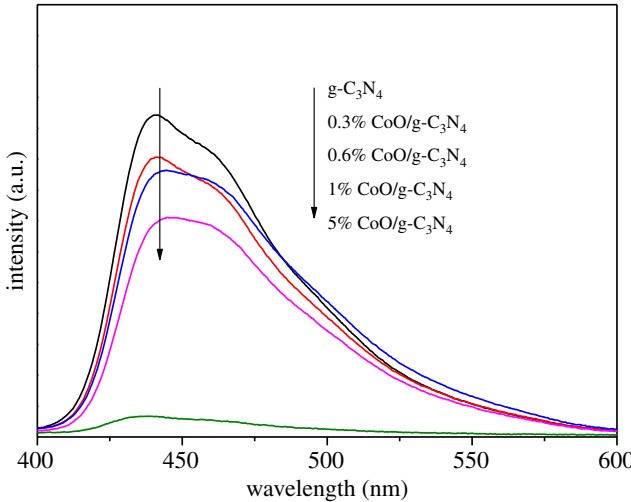

**Figure 5.** PL spectra ($\lambda_{ex}$ = 370 nm) for the prepared g-C$_3$N$_4$, 0.3% CoO/g-C$_3$N$_4$, 0.6% CoO/g-C$_3$N$_4$, 1% CoO/g-C$_3$N$_4$ and 5% CoO/ g-C$_3$N$_4$ composite.

heterojunctions with a faster separation efficiency of photogenerated electron–hole pairs. The PL analysis is usually carried out to investigate the transfer, migration and recombination of photogenerated electron–hole pairs of the photocatalyst [26]. The PL spectra of as-prepared g-C$_3$N$_4$ and CoO/g-C$_3$N$_4$ composites with excitation wavelength of 370 nm are demonstrated in figure 5. CoO/g-C$_3$N$_4$ composite exhibits a much weaker emission profile with the CoO loading content increasing in comparison with g-C$_3$N$_4$, indicating that the recombination rate of the photogenerated charge carrier is greatly restrained and there is a faster photoelectron transfer between the hybrid of CoO and g-C$_3$N$_4$.

The photocatalytic H$_2$ evolution performance of the prepared CoO/g-C$_3$N$_4$ composite with different CoO content is measured using Pt as a co-catalyst and the results are shown in figure 6. In figure 6*a*, it can be found that the photocatalytic H$_2$ evolution amount for CoO/g-C$_3$N$_4$ composite with 0, 0.3, 0.6, 1, 5 and 100 wt% Co loading content is recorded to be 14.79, 17.19, 23.25, 13.02, 1.90 and 0.019 mmol g$^{-1}$ after 5 h, respectively. The photocatalytic H$_2$ evolution amount increases as the Co content increases from 0 to 0.6 wt% and then exhibits a decrease with a higher Co content. This decrease is possible due to excessive CoO aggregation and the decrease in g-C$_3$N$_4$ surface active sites. The 0.6 wt% CoO/ g-C$_3$N$_4$ composite exhibits the best photocatalytic performance with an average hydrogen evolution rate of 4.65 mmol h$^{-1}$ g$^{-1}$, which is about 57% higher than that of pure g-C$_3$N$_4$. Compared with the reported 0.5 wt% CoO/g-C$_3$N$_4$ (0.65 mmol h$^{-1}$ g$^{-1}$) [35], 30 wt% CoO/g-C$_3$N$_4$ (2.51 μmol h$^{-1}$) [26] and 10 wt% CoO/g-C$_3$N$_4$ (0.46 μmol h$^{-1}$) [25], photocatalytic hydrogen evolution performance of

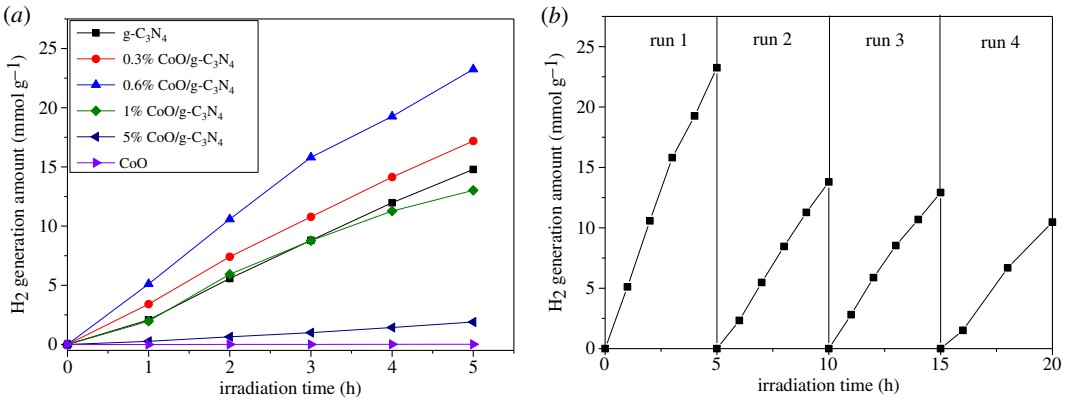

**Figure 6.** (a) The photocatalytic $H_2$ evolution amount of the samples; (b) recyclability of 0.6 wt% $CoO/g-C_3N_4$ photocatalyst for the photocatalytic $H_2$ evolution, with 10 vol% TEOA, 1.5 wt% Pt.

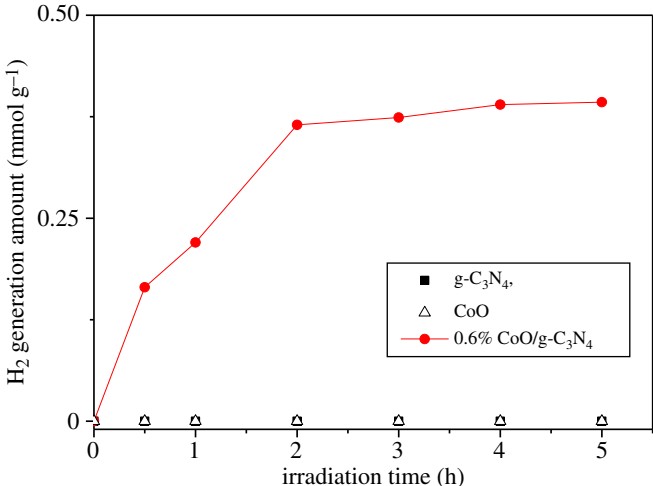

**Figure 7.** $H_2$ evolutions from pure water with $g-C_3N_4$, CoO and 0.6 wt% $CoO/g-C_3N_4$.

$CoO/g-C_3N_4$ composite could be improved by the rotation–evaporation and thermal annealing methods. Figure 6$b$ shows the photocatalytic stability of hydrogen evolution for the 0.6 wt% $CoO/g-C_3N_4$ sample is carried out by four cycling experiments under the same condition. The photocatalytic $H_2$ evolution activity of 0.6% $CoO/g-C_3N_4$ exhibits favourable stability for the four recycling runs.

In the following experiments, as 0.6 wt% $CoO/g-C_3N_4$ exhibits the superior photocatalytic $H_2$ evolution activity, its $H_2$ evolution performance for splitting pure water is investigated and the results are shown in figure 7. It is found that 0.6 wt% $CoO/g-C_3N_4$ can split pure water to generate $H_2$ without sacrificial agent and noble metal Pt, while the pure $g-C_3N_4$ and bulk CoO exhibit negligible photocatalytic activity towards $H_2$ generation under the same reaction condition. $H_2O_2$ is more easily generated than $O_2$, which is attributed to the more positive $H_2O_2/H_2O$ (1.78 V versus RHE) than $O_2/H_2O$ (1.23 V versus RHE) [26]. The drawbacks of rapid rate of photogenerated electron–hole pairs and severe poisoning by $H_2O_2$ generated during the reaction of $g-C_3N_4$ are the main reasons for the inactivation [13]. The photocatalytic $H_2$ evolution amount of 0.6% $CoO/g-C_3N_4$ reaches 0.39 mmol g$^{-1}$ and the photocatalytic $H_2$ evolution rate is very slow after 2 h under light irradiation. Based on these results, it can be safely concluded that $g-C_3N_4$ doped with 0.6 wt% CoO could effectively separate the photogenerated electron–hole pairs to generate $H_2$ from pure water splitting; however, it is likely subject to being greatly poisoned by $H_2O_2$ generation during the photocatalytic reaction to cause rapid inactivation.

The photogenerated charge transfer and separation properties in the 0.6 wt% $CoO/g-C_3N_4$ sample are further studied by photoelectrochemical measurements. EIS and photocurrents are carried out to obtain the photogenerated charge separation and transfer properties. Figure 8$a$ shows the transient photocurrent responses for $g-C_3N_4$ and 0.6% $CoO/g-C_3N_4$ samples with an interval light on/off cycle

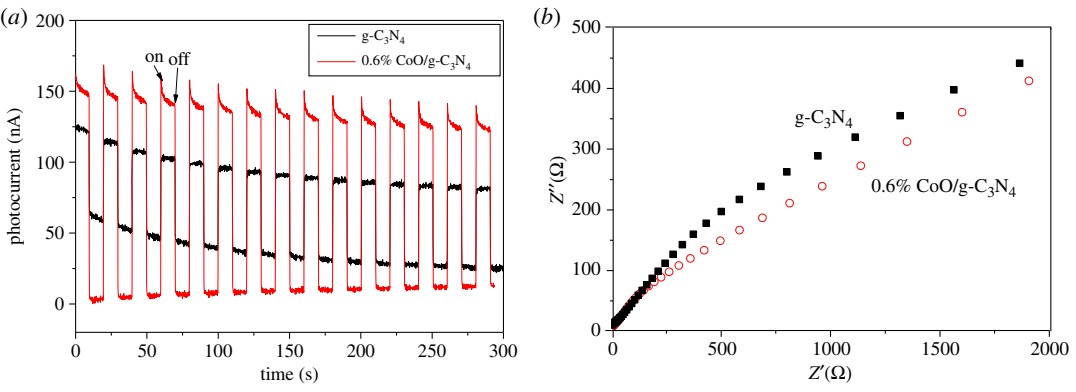

**Figure 8.** (a) Transient photocurrents, (b) electrochemical impedance spectra of g-C$_3$N$_4$ and 0.6% CoO/g-C$_3$N$_4$ electrodes at 0.3 and −0.4 V versus Ag/AgCl.

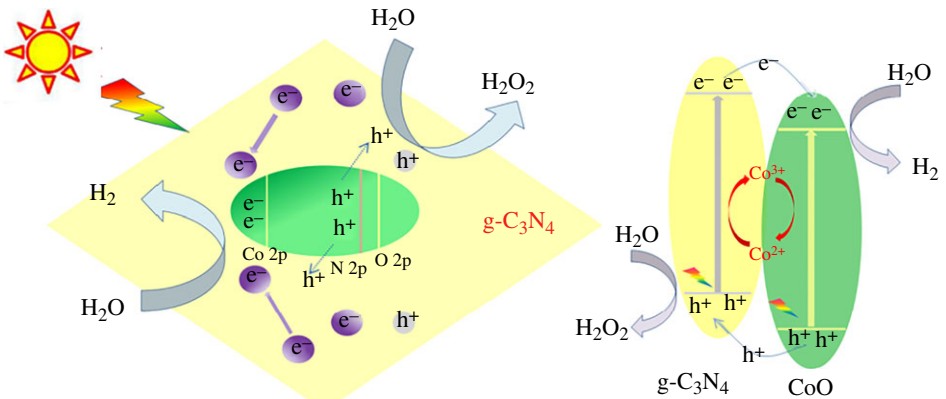

**Figure 9.** Schematic illustrations of the proposed photocatalytic H$_2$ evolution mechanism for overall water splitting over the CoO/ g-C$_3$N$_4$ hybrid catalyst.

mode. It can be clearly observed that the transient photocurrent response of 0.6% CoO/g-C$_3$N$_4$ composite is higher than that of the pure g-C$_3$N$_4$ sample, suggesting that the interfical electron transfer and electron–hole separation process between CoO and g-C$_3$N$_4$ is highly proven. The EIS measurements have been carried out to evaluate the photogenerated electron transfer in CoO/g-C$_3$N$_4$. Figure 8b presents the experimental Nyquist impedance plots for g-C$_3$N$_4$ and 0.6% CoO/g-C$_3$N$_4$ samples in the dark. The Nyquist plot of 0.6% CoO/g-C$_3$N$_4$ suggests an apparently smaller arc diameter than that of g-C$_3$N$_4$, indicating that the 0.6% CoO/g-C$_3$N$_4$ system can efficiently migrate interfacial charge and separate the photogenerated electron–hole pairs at the heterojunction interface across the electrode/ electrolyte in agreement with the results of PL, contributing to the enhancement of photocatalytic hydrogen evolution activity [26].

Based on the above experimental results, a possible mechanism of photocatalytic H$_2$ evolution for the CoO/g-C$_3$N$_4$ hybrid system is proposed, as shown in figure 9. First, the electron–hole pairs [38,39] are generated in the conduction band [40] and valence band of g-C$_3$N$_4$ by light irradiation. Then, the photogenerated holes in the valence band of g-C$_3$N$_4$ will react with H$_2$O to generate H$_2$O$_2$. In contrary, the photogenerated electrons further transfer from the conduction band of g-C$_3$N$_4$ to the surface of CoO nanoparticles, which function as reduction catalysts to catalyse the reduction in protons (H$^+$) to H$_2$. Therefore, the separation of electron–hole pairs and the charge transfer can effectively enhance the photocatalytic H$_2$ evolution activity from overall water splitting for the CoO/ g-C$_3$N$_4$ heterojunction photocatalyst.

## 4. Conclusion

The CoO/g-C$_3$N$_4$ hybrid catalysts with different CoO loading contents are facilely prepared to study the photocatalytic H$_2$ evolution activity. CoO/g-C$_3$N$_4$ composite with 0.6 wt% Co loading shows the highest

photocatalytic activity for $H_2$ evolution amount of $23.25\,mmol\,g^{-1}$ after $5\,h$, which is 57% higher than that of g-$C_3N_4$. The remarkably enhanced photocatalytic performance for $H_2$ evolution of CoO/g-$C_3N_4$ composite is mainly due to the faster transfer of interfacial charge and more effective separation of electron–hole pairs. The photocatalytic $H_2$ evolution amount of 0.6% CoO/g-$C_3N_4$ reaches $0.39\,mmol\,g^{-1}$ by overall water splitting without sacrificial agent and noble metal. But the photocatalytic $H_2$ evolution rate of 0.6% CoO/g-$C_3N_4$ is very slow after $2\,h$ because it is easily poisoned by $H_2O_2$ generation during the photocatalytic reaction to cause rapid inactivation. In future work, CoO/g-$C_3N_4$ material with the stability of photocatalytic $H_2$ evolution activity will be further designed to prevent $H_2O_2$ poisoning.

Data accessibility. Data available from the Dryad Digital Repository at: https://doi.org/10.5061/dryad.cs0sj00 [31].
Authors' contributions. X.L. and L.W. designed the study. L.L. and Q.Z. prepared all the samples for analysis. L.C., Y.W. and X.T carried out the statistical analyses. H.H. interpreted the results. All the authors gave their final approval for publication.
Competing interests. There are no conflicts to declare.
Funding. This work was supported by Guangdong Provincial Key Laboratory of New and Renewable Energy Research and Development (grant no. Y807s21001) and Research Foundation for Talented Scholars (grant no. 1856012).
Disclaimer. We have read the information for authors and publish policy. We all agree with the policy and prepare the manuscript in accordance with the guidance.

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
