## [Reviewer comments · Royal Society Open Science]

Review History

RSOS-190433.R0 (Original submission)

Review form: Reviewer 1 (Ummadisetti Chinnarajesh)

Is the manuscript scientifically sound in its present form?

No

Are the interpretations and conclusions justified by the results?

Yes

Is the language acceptable?

Yes

Is it clear how to access all supporting data?

No

Do you have any ethical concerns with this paper?

No

Have you any concerns about statistical analyses in this paper?

No

Recommendation?

Reject

Comments to the Author(s)

Journal: Royal Society Open Science

Manuscript ID RSOS-190433

Liu et. al reported the "In-situ growing of CoO nanoparticles on g-C₃N₄ composites by facile thermal annealing method with highly improved photocatalytic activity for hydrogen evolution." However, the synthesis of CoO@g-C₃N₄ nanocomposites thermal annealing method and their photocatalytic potential for water splitting are reported in the literature (see Ref. Han et al. *Inorg. Chem. Front.*, 2017, 4, 1691-1696; Guo et al. *Appl. Catal. B: Environ.*, 226, 2018, 412-420; Wang et al., *Inorg. Chem. Commun.*, 92, 2018, 14-17). The present work does not contribute any new findings to the readers of material science/nanocatalysis areas. Moreover, Royal Society Open Science journal publishes high-quality original research, in this sense, the novelty of present work is not sufficient to warrant a publication in RSOS journal.

Major comments

1. The introduction of this manuscript is missing the important information on CoO as a photocatalyst for water splitting, and H₂O₂ poisoning of catalyst surface and challenges to overcome the drawbacks with citing the relevant Ref.'s for example, Xu et al. *Angew. Chem. Int. Ed.* 2019, 58, 3032 -3036, DOI: 10.1002/anie.201807332. Shi et al. *ACS Appl. Mater. Interfaces*, 2017, 20585-20593, DOI: 10.1021/acsami.7b04286; J. Mater. Chem. A, 2017, 19800-19807, 10.1039/C7TA06077G; Neatu, *Chem. Commun.*, 2014,14643-14646, DOI: 10.1039/c4cc05931j;
2. Authors mentioned (presumed) in the Page 3 Line 14,... "Therefore, the combination of g-C₃N₄ and CoO could achieve..... sacrificial agent. However, the combination of g-C₃N₄ and CoO NPs has already known to be improved photocatalysts for water splitting in the literature. (see Ref. Han et al. *Inorg. Chem. Front.*, 2017, 4, 1691-1696; Guo et al. *Applied Catalysis B: Environmental*, 226, 2018, 412-420; Wang et al., *Inorganic Chemistry Communications*, 92, 2018, 14-17). This phrase should be rewritten by including the above Ref.'s
3. Thermal annealing method for preparation of CoO@g-C₃N₄ composites is known in the literature (Ref, Han et al. *Inorg. Chem. Front.*, 2017, 4, 1691-1696). Authors should justify how their approach is different or improved process than the reported Ref.
4. The main focus of present work is to explore the catalytic potential of CoO/g-C₃N₄ composites without Pt co-catalyst. The present draft shows more emphasis on Pt co-catalyst along with CoO/g-C₃N₄ in Fig 6a. Authors should move these controls into the Supp Information, and change the samples notation, for example, g-C₃N₄@Pt, 0.3%CoO/g-C₃N₄@Pt.....etc inset of Fig 6a..Include controlled catalysis of CoO alone and various wt% of CoO on g-C₃N₄ composites for water splitting in the main manuscript.
5. The photocurrent studies of the composites in the presence of light and dark need to be included in the manuscript.
6. The XPS characterization of recovered 0.6 wt% CoO/g-C₃N₄ catalyst need to be included to see the changes on surfaces.

Minor/technical corrections

7. The scale bars and font size and colors (change black to white) of notations a, b, c were not clear in Fig 2. Move three Figures in one row by cropping the size of TEM images.

8. Include the detailed sample notations in the Figures, for example In Fig 6a, Pt needs to be included and it is not clear whether Pt is included or not in the Fig 6b analysis.

9. The notations in Fig 4 and 5 are not clear to the readers. Include a, b, c...etc. on the curves with appropriate color on the sample codes.

Review form: Reviewer 2

Is the manuscript scientifically sound in its present form?

No

Are the interpretations and conclusions justified by the results?

No

Is the language acceptable?

Yes

Is it clear how to access all supporting data?

Yes

Do you have any ethical concerns with this paper?

No

Have you any concerns about statistical analyses in this paper?

No

Recommendation?

Major revision is needed (please make suggestions in comments)

Comments to the Author(s)

The novelty of the work is not reflected in the introduction. Additionally, the the result and discussion for hydrogen generation is not compared with respect to the bare CoO to highlight or to show the novelty of their work. However, the paper can be published after careful observation the reply of the major revision.

1. Why the peak (532.2) of 5 wt% CoO/g-C₃N₄ is shifted to the lower binding energy position? It is necessary to explain the trend of binding energy to the lower energy level with the increasing the CoO wt%.

2. The author need to give information of CoO/C₃N₄ composite prepared by the other groups?

3. Is the synthesis process of CoO, C₃N₄ and CoO/C₃N₄ is novel or other researcher already prepared the materials by the same method?

4. The author need to give the data of photocatalytic hydrogen evolution the bare CoO catalyst. It is necessary to compare it(bare CoO) with the CoO/C₃N₄ in Figure 6a (hydrogen generation), Figure 7 (hydrogen generation), and in Figure 8 (impedance).

Review form: Reviewer 3

Is the manuscript scientifically sound in its present form?

No

Are the interpretations and conclusions justified by the results?

Yes

Is the language acceptable?

Yes

Is it clear how to access all supporting data?

No

Do you have any ethical concerns with this paper?

No

Have you any concerns about statistical analyses in this paper?

No

Recommendation?

Major revision is needed (please make suggestions in comments)

Comments to the Author(s)

In this work, CoO nanoparticles are in-situ growing on the g-C₃N₄ to prepare CoO/g-C₃N₄ composite photocatalyst by facile thermal annealing method under nitrogen atmosphere. The enhancement mechanism of photocatalytic overall water splitting for hydrogen evolution of as-synthesized CoO/g-C₃N₄ nanocomposite is also discussed. There are still some shortcomings in the paper. I believe the paper may be accepted for publication after carefully addressing the following points.

1. CoO nanoparticles are in-situ growing on the g-C₃N₄ in this paper. What is the novelty or advantage over other literatures? The reason or characteristic for this method need to be emphasized in Introduction.
2. The results from this paper could be compared with the data from previous literatures.
3. Some expressions need to improved, such as "can't generate H₂ under"
4. The photocatalytic mechanisms of composites for water need to be discussed further, some reference could be referred. *Ceramics International*, 44 (2018), 1711-1718, *Journal of Membrane Science*, 520 (2016), 281-293; *Ceramics International*, 42 (2016), 15012-15022.
5. In the introduction, you should note the problems with traditional CoO/g-C₃N₄ combination methods, which can make the reader know what is unique about your work.
6. There are many previous works about the in-situ synthesis of nanoparticles, such as DOI:10.1016/j.electacta.2018.10.039, 10.1016/j.snb.2019.02.026 and so on. Is there improvement for in-situ synthesis? These need to be explained in Introduction.

Decision letter (RSOS-190433.R0)

24-Apr-2019

Dear Dr Liu:

Title: In-situ growing of CoO nanoparticles on g-C₃N₄ composites with highly improved photocatalytic activity for hydrogen evolution
Manuscript ID: RSOS-190433

The editor assigned to your manuscript has now received comments from reviewers. We would like you to revise your paper in accordance with the referee and Subject Editor suggestions which can be found below (not including confidential reports to the Editor). Please note this decision does not guarantee eventual acceptance.

Please submit your revised paper before 17-May-2019. Please note that the revision deadline will expire at 00.00am on this date. If we do not hear from you within this time then it will be assumed that the paper has been withdrawn. In exceptional circumstances, extensions may be possible if agreed with the Editorial Office in advance. We do not allow multiple rounds of revision so we urge you to make every effort to fully address all of the comments at this stage. If deemed necessary by the Editors, your manuscript will be sent back to one or more of the original reviewers for assessment. If the original reviewers are not available we may invite new reviewers.

Please also include the following statements alongside the other end statements. As we cannot publish your manuscript without these end statements included, if you feel that a given heading is not relevant to your paper, please nevertheless include the heading and explicitly state that it is not relevant to your work.

- Acknowledgements

Yours sincerely,
Dr Laura Smith

Publishing Editor, Journals

RSC Associate Editor:

Comments to the Author:

Please address the reviewers' comments in full, in particular addressing their concerns about the advance in scientific knowledge presented in this work.

RSC Subject Editor:

Comments to the Author:

(There are no comments.)

Reviewers' Comments to Author:

Reviewer: 1

Comments to the Author(s)

Journal: Royal Society Open Science

Manuscript ID RSOS-190433

Liu et. al reported the "In-situ growing of CoO nanoparticles on g-C₃N₄ composites by facile thermal annealing method with highly improved photocatalytic activity for hydrogen evolution." However, the synthesis of CoO@g-C₃N₄ nanocomposites thermal annealing method and their photocatalytic potential for water splitting are reported in the literature (see Ref. Han et al. *Inorg. Chem. Front.*, 2017, 4, 1691-1696; Guo et al. *Appl. Catal. B: Environ.*, 226, 2018, 412-420; Wang et al., *Inorg. Chem. Commun.*, 92, 2018, 14-17). The present work does not contribute any new findings to the readers of material science/nanocatalysis areas. Moreover, Royal Society Open Science journal publishes high-quality original research, in this sense, the novelty of present work is not sufficient to warrant a publication in RSOS journal.

Major comments

1. The introduction of this manuscript is missing the important information on CoO as a photocatalyst for water splitting, and H₂O₂ poisoning of catalyst surface and challenges to overcome the drawbacks with citing the relevant Ref.'s for example, Xu et al. *Angew. Chem. Int. Ed.* 2019, 58, 3032 -3036, DOI: 10.1002/anie.201807332. Shi et al. *ACS Appl. Mater. Interfaces*, 2017, 20585-20593, DOI: 10.1021/acsami.7b04286; J. Mater. Chem. A, 2017, 19800-19807, 10.1039/C7TA06077G; Neatu, *Chem. Commun.*, 2014,14643-14646, DOI: 10.1039/c4cc05931j;
2. Authors mentioned (presumed) in the Page 3 Line 14,... "Therefore, the combination of g-C₃N₄ and CoO could achieve..... sacrificial agent. However, the combination of g-C₃N₄ and CoO NPs has already known to be improved photocatalysts for water splitting in the literature. (see Ref. Han et al. *Inorg. Chem. Front.*, 2017, 4, 1691-1696; Guo et al. *Applied Catalysis B: Environmental*,

226, 2018, 412-420; Wang et al., *Inorganic Chemistry Communications*, 92, 2018, 14-17). This phrase should be rewritten by including the above Ref.'s

3. Thermal annealing method for preparation of CoO@g-C₃N₄ composites is known in the literature (Ref, Han et al. *Inorg. Chem. Front.*, 2017, 4, 1691-1696). Authors should justify how their approach is different or improved process than the reported Ref.

4. The main focus of present work is to explore the catalytic potential of CoO/g-C₃N₄ composites without Pt co-catalyst. The present draft shows more emphasis on Pt co-catalyst along with CoO/g-C₃N₄ in Fig 6a. Authors should move these controls into the Supp Information, and change the samples notation, for example, g-C₃N₄@Pt, 0.3%CoO/g-C₃N₄@Pt.....etc inset of Fig 6a..Include controlled catalysis of CoO alone and various wt% of CoO on g-C₃N₄ composites for water splitting in the main manuscript.

5. The photocurrent studies of the composites in the presence of light and dark need to be included in the manuscript.

6. The XPS characterization of recovered 0.6 wt% CoO/g-C₃N₄ catalyst need to be included to see the changes on surfaces.

Minor/technical corrections

7. The scale bars and font size and colors (change black to white) of notations a, b, c were not clear in Fig 2. Move three Figures in one row by cropping the size of TEM images.

8. Include the detailed sample notations in the Figures, for example In Fig 6a, Pt needs to be included and it is not clear whether Pt is included or not in the Fig 6b analysis.

9. The notations in Fig 4 and 5 are not clear to the readers. Include a, b, c...etc. on the curves with appropriate color on the sample codes.

Reviewer: 2

Comments to the Author(s)

The novelty of the work is not reflected in the introduction. Additionally, the the result and discussion for hydrogen generation is not compared with respect to the bare CoO to highlight or to show the novelty of their work. However, the paper can be published after careful observation the reply of the major revision.

1. Why the peak (532.2) of 5 wt% CoO/g-C₃N₄ is shifted to the lower binding energy position? It is necessary to explain the trend of binding energy to the lower energy level with the increasing the CoO wt%.

2. The author need to give information of CoO/C₃N₄ composite prepared by the other groups?

3. Is the synthesis process of CoO, C₃N₄ and CoO/C₃N₄ is novel or other researcher already prepared the materials by the same method?

4. The author need to give the data of photocatalytic hydrogen evolution the bare CoO catalyst. It is necessary to compare it(bare CoO) with the CoO/C₃N₄ in Figure 6a (hydrogen generation), Figure 7 (hydrogen generation), and in Figure 8 (impedance).

Reviewer: 3

Comments to the Author(s)

In this work, CoO nanoparticles are in-situ growing on the g-C₃N₄ to prepare CoO/g-C₃N₄ composite photocatalyst by facile thermal annealing method under nitrogen atmosphere. The

enhancement mechanism of photocatalytic overall water splitting for hydrogen evolution of as-synthesized CoO/g-C₃N₄ nanocomposite is also discussed. There are still some shortcomings in the paper. I believe the paper may be accepted for publication after carefully addressing the following points.

1. CoO nanoparticles are in-situ growing on the g-C₃N₄ in this paper. What is the novelty or advantage over other literatures? The reason or characteristic for this method need to be emphasized in Introduction.
2. The results from this paper could be compared with the data from previous literatures.
3. Some expressions need to improved, such as "can't generate H₂ under"
4. The photocatalytic mechanisms of composites for water need to be discussed further, some reference could be referred. *Ceramics International*, 44 (2018), 1711-1718, *Journal of Membrane Science*, 520 (2016), 281-293; *Ceramics International*, 42 (2016), 15012-15022.
5. In the introduction, you should note the problems with traditional CoO/g-C₃N₄ combination methods, which can make the reader know what is unique about your work.
6. There are many previous works about the in-situ synthesis of nanoparticles, such as DOI:10.1016/j.electacta.2018.10.039, 10.1016/j.snb.2019.02.026 and so on. Is there improvement for in-situ synthesis? These need to be explained in Introduction.

Author's Response to Decision Letter for (RSOS-190433.R0)

See Appendix A.

RSOS-190433.R1 (Revision)

Review form: Reviewer 3

Is the manuscript scientifically sound in its present form?

Yes

Are the interpretations and conclusions justified by the results?

Yes

Is the language acceptable?

Yes

Is it clear how to access all supporting data?

Yes

Do you have any ethical concerns with this paper?

No

Have you any concerns about statistical analyses in this paper?

No

Recommendation?

Accept as is

Comments to the Author(s)

I agree to accept this article which meeting the publication requirements.

Decision letter (RSOS-190433.R1)

11-Jun-2019

Dear Dr Liu:

Title: In-situ growing of CoO nanoparticles on g-C₃N₄ composites with highly improved photocatalytic activity for hydrogen evolution

Manuscript ID: RSOS-190433.R1

It is a pleasure to accept your manuscript in its current form for publication in Royal Society Open Science. The chemistry content of Royal Society Open Science is published in collaboration with the Royal Society of Chemistry.

RSC Associate Editor:
Comments to the Author:
(There are no comments.)

RSC Subject Editor:
Comments to the Author:
(There are no comments.)

Reviewer(s)' Comments to Author:

Reviewer: 3

Comments to the Author(s)

I agree to accept this article which meeting the publication requirements.

Appendix A

Response to Reviewer's comments

Reviewer: 1

1. The introduction of this manuscript is missing the important information on CoO as a photocatalyst for water splitting, and H₂O₂ poisoning of catalyst surface and challenges to overcome the drawbacks with citing the relevant Ref.'s for example, Xu et al. *Angew. Chem. Int. Ed.* 2019, 58, 3032–3036, DOI: 10.1002/anie.201807332. Shi et al. *ACS Appl. Mater. Interfaces*, 2017, 20585-20593, DOI: 10.1021/acsami.7b04286; *J. Mater. Chem. A*, 2017, 19800-19807, 10.1039/C7TA06077G; Neatu, *Chem. Commun.*, 2014, 14643-14646, DOI: 10.1039/c4cc05931j;

Response to Reviewer comment No. 1: According to the advice of reviewer, the reference [*Angew. Chem. Int. Ed.* 2019, 58, 3032-3036] about CoO as a photocatalyst for water splitting have been cited in the introduction section of revised manuscript. The references [*ACS Appl. Mater. Interfaces*, 2017, 20585-20593; *J. Mater. Chem. A*, 2017, 19800-19807; *Chem. Commun.*, 2014, 14643-14646] about have been cited in the revised manuscript. Corresponding changes have been in the revised manuscript at line 66-69 as follow.

2. Authors mentioned (presumed) in the Page 3 Line 14,... “Therefore, the combination of g-C₃N₄ and CoO could achieve..... sacrificial agent. However, the combination of g-C₃N₄ and CoO NPs has already known to be improved photocatalysts for water splitting in the literature. (see Ref. Han et al. *Inorg. Chem. Front.*, 2017, 4, 1691–1696; Guo et al. *Applied Catalysis B: Environmental*, 226, 2018, 412-420; Wang et al., *Inorganic Chemistry Communications*, 92, 2018, 14-17). This phrase should be rewritten by including the above Ref.'s

Response to Reviewer comment No. 2: The phrase “It is reported that the combination of g-C₃N₄ and CoO can be improved photocatalysts for water splitting.” has been rewritten by the references [*Inorg. Chem. Front.*, 2017, 4, 1691-1696; *Applied Catalysis B: Environmental*, 226, 2018, 412-420; *Inorganic Chemistry Communications*, 92, 2018, 14-17] in the revised manuscript. Corresponding changes have been in the revised manuscript at line 69-74 as follow.

3. Thermal annealing method for preparation of CoO@g-C₃N₄ composites is known in the literature (Ref, Han et al. Inorg. Chem. Front., 2017, 4, 1691-1696). Authors should justify how their approach is different or improved process than the reported Ref.

Response to Reviewer comment No.3: It is well known that the nanoparticles could be highly dispersed by rotation-evaporation method. Therefore, photocatalytic hydrogen generation performance of CoO/g-C₃N₄ composite will be improved by facile thermal annealing and rotation-evaporation method compared with the reported references. Corresponding changes have been in the revised manuscript at line 72 as follow.

4. The main focus of present work is to explore the catalytic potential of CoO/g-C₃N₄ composites without Pt co-catalyst. The present draft shows more emphasis on Pt co-catalyst along with CoO/g-C₃N₄ in Fig 6a. Authors should move these controls into the Supp Information, and change the samples notation, for example, g-C₃N₄@Pt, 0.3%CoO/g-C₃N₄@Pt.....etc inset of Fog 6a. Include controlled catalysis of CoO alone and various wt% of CoO on g-C₃N₄ composites for water splitting in the main manuscript.

Response to Reviewer comment No. 4: In this paper, CoO/g-C₃N₄ composites with Pt co-catalyst are investigated to explore CoO/g-C₃N₄ composite with well-defined characteristics for overall water splitting. In the caption of figure 6a, the reaction condition with 1.5wt% Pt has been specified for the reader's better understanding. The photocatalytic overall water-splitting performance of pure CoO has been studied in the Figure 7. It is found that 0.6wt% CoO/g-C₃N₄ can split pure water to generate H₂ without sacrificial agent and noble metal Pt, while the pure g-C₃N₄ and bulk CoO exhibit negligible photocatalytic activity towards H₂ generation under the same reaction condition. Corresponding changes have been in the revised manuscript.

5. The photocurrent studies of the composites in the presence of light and dark need to be included in the manuscript.

Response to Reviewer comment No. 5: The photocurrent studies of the composites in

the presence of light and dark have been studied in the manuscript, as shown in figure 8(a). Photocurrent measurements are conducted on a conventional three-electrode system in an alternating voltage of 0.02 V under chopped illumination with 40 s light on and off (dark) cycles.

6. The XPS characterization of recovered 0.6 wt% CoO/g-C₃N₄ catalyst need to be included to see the changes on surfaces.

Response to Reviewer comment No. 6: The XPS characterization of recovered 0.6 wt% CoO/g-C₃N₄ catalyst has been carried on to see the changes on surfaces. The following figure R1 shows XPS profiles of O 1s and Co 2p of 0.6 wt% CoO/g-C₃N₄ and recovered 0.6 wt% CoO/g-C₃N₄ samples. After being recovered, the signal of Co 2p has no significant change. The O 1s spectra with two peaks at about 529 eV and 532 eV are shown in figure R1. The binding energy at 529 eV is ascribed to the Co-O bond in the CoO phase, while the strong peak at about 532 eV corresponds to the Co-O-C bond, indicating that a strong interaction exists between CoO and g-C₃N₄. It can be seen that the signal of Co-O-C bond in the recovered 0.6 wt% CoO/g-C₃N₄ catalyst becomes much bigger than pure 0.6 wt% CoO/g-C₃N₄ because of the change of electronic state of adsorbed oxygen species by formed H₂O₂.

Figure R1 XPS profiles of O 1s and Co 2p of 0.6 wt% CoO/g-C₃N₄ and recovered 0.6 wt% CoO/g-C₃N₄ samples.

7. The scale bars and font size and colors (change black to white) of notations a, b, c were not clear in Fig 2. Move three Figures in one row by cropping the size of TEM images.

Response to Reviewer comment No. 7: In the revised manuscript, the scale bars and font size and colors of notations a, b, c have been changed black to white and the size of TEM images have been dropped in one row in the figure 2.

8. Include the detailed sample notations in the Figures, for example In Fig 6a, Pt needs to be included and it is not clear whether Pt is included or not in the Fig 6b analysis.

Response to Reviewer comment No. 8: All the experiments in figure 6a and 6b are measured in the 10 vol% TEOA solution with 1.5 wt% Pt. Corresponding changes have been in the caption of figure 6 in revised manuscript.

9. The notations in Fig 4 and 5 are not clear to the readers. Include a, b, c...etc. on the curves with appropriate color on the sample codes.

Response to Reviewer comment No. 9: In figure 4 and figure 5, notations of the samples are correspondence with the order of lines along the arrow.

Reviewer: 2

1. Why the peak (532.2) of 5 wt% CoO/g-C₃N₄ is shifted to the lower binding energy position? It is necessary to explain the trend of binding energy to the lower energy level with the increasing the CoO wt%.

Response to Reviewer comment No. 1: The strong peak at about 532 eV corresponds to the Co-O-C bond, while the binding energy at 529 eV is ascribed to the Co-O bond in the CoO phase.³⁵ The peak (532.2 eV) of 5 wt% CoO/g-C₃N₄ shifted to the lower binding energy position is possible due to the stronger Co-O bond than the Co-O-C

bond in the composites with the increasing the CoO wt%.

2. The author need to give information of CoO/C₃N₄ composite prepared by the other groups?

Response to Reviewer comment No. 2: CoO/C₃N₄ composites prepared by the other groups [ref. 22-27] have been provided in the introduction section. But poor stability of the CoO catalyst is caused by H₂O₂ poisoning to hinder its further development.²²⁻²⁴ It is still a challenge to seek suitable structure of CoO based catalyst with high stability. It is reported that the combination of g-C₃N₄ and CoO can be improved photocatalysts for water splitting.²⁵⁻²⁷

3. Is the synthesis process of CoO, C₃N₄ and CoO/C₃N₄ is novel or other researcher already prepared the materials by the same method?

Response to Reviewer comment No. 3: CoO/C₃N₄ composite prepared by thermal annealing method has been reported by other literatures²⁵⁻²⁷. As the nanoparticles could be highly dispersed by vacuum rotation-evaporation method, in this paper, CoO/g-C₃N₄ composite is improved by vacuum rotation-evaporation and thermal annealing method compared with these reported references.

4. The author need to give the data of photocatalytic hydrogen evolution the bare CoO catalyst. It is necessary to compare it (bare CoO) with the CoO/C₃N₄ in Figure 6a (hydrogen generation), Figure 7 (hydrogen generation), and in Figure 8 (impedance).

Response to Reviewer comment No. 4: The photocatalytic hydrogen evolution performance of bare CoO has been provided in the Figure 6a and Figure 7. In figure 6(a), it can be found that the photocatalytic H₂ evolution amount for CoO/g-C₃N₄ composite with 0, 0.3, 0.6, 1, 5 and 100 wt% Co loading content is recorded to be 14.79, 17.19, 23.25, 13.02, 1.90 and 0.019 mmol g⁻¹ after 5h, respectively. It is found

that 0.6wt% CoO/g-C₃N₄ can split pure water to generate H₂ without sacrificial agent and noble metal Pt, while the pure g-C₃N₄ and bulk CoO exhibit negligible photocatalytic activity towards H₂ generation under the same reaction condition.

Reviewer: 3

1. CoO nanoparticles are in-situ growing on the g-C₃N₄ in this paper. What is the novelty or advantage over other literatures? The reason or characteristic for this method need to be emphasized in Introduction.

Response to Reviewer comment No. 1: It is still a challenge to seek suitable structure of CoO based catalyst with high activity and stability. CoO/C₃N₄ composite prepared by thermal annealing method has been reported by other literatures. As the nanoparticles could be highly dispersed by vacuum rotation-evaporation method, in this paper, CoO/g-C₃N₄ composite is improved by vacuum rotation-evaporation and thermal annealing method compared with these reported references. Corresponding changes have been in the revised manuscript at line 68-71 as follow.

2. The results from this paper could be compared with the data from previous literatures.

Response to Reviewer comment No. 2: Compared with the reported 0.5 wt% CoO/g-C₃N₄ (0.65 mmol h⁻¹ g⁻¹),³⁴ 30 wt% CoO/g-C₃N₄ (2.51 μmol h⁻¹)²⁶ and 10 wt% CoO/g-C₃N₄ (0.46 μmol h⁻¹),²⁵ photocatalytic hydrogen evolution performance of CoO/g-C₃N₄ composite could be improved by rotation-evaporation and thermal annealing method. Corresponding changes have been in the revised manuscript at line 232-236 as follow.

3. Some expressions need to improved, such as “can’t generate H₂ under”.

Response to Reviewer comment No.3: The expressions have been modified. It is

found that 0.6wt% CoO/g-C₃N₄ can split pure water to generate H₂ without sacrificial agent and noble metal Pt, while the pure g-C₃N₄ and bulk CoO exhibit negligible photocatalytic activity towards H₂ generation under the same reaction condition. Corresponding changes have been in the revised manuscript at line 244-245 as follow.

4. The photocatalytic mechanisms of composites for water need to be discussed further, some reference could be referred. *Ceramics International*, 44 (2018), 1711-1718, *Journal of Membrane Science*, 520 (2016), 281-293; *Ceramics International*, 42 (2016), 15012-15022.

Response to Reviewer comment No. 4: According to the advice of reviewer, the references [*Ceramics International*, 44 (2018), 1711-1718, *Journal of Membrane Science*, 520 (2016), 281-293; *Ceramics International*, 42 (2016), 15012-15022] have been cited in the revised manuscript. Corresponding changes have been in the revised manuscript at line 293 as follow.

5. In the introduction, you should note the problems with traditional CoO/g-C₃N₄ combination methods, which can make the reader know what is unique about your work.

Response to Reviewer comment No. 5: The problems with traditional CoO based materials have been noted in the introduction section in the revised manuscript. CoO with efficient photo-induced electrons separation can be used as an effectively co-catalyst to improve the photocatalytic water splitting activity for hydrogen evolution. It is reported that the combination of g-C₃N₄ and CoO can be improved photocatalysts for water splitting. But poor stability of the CoO catalyst is caused by H₂O₂ poisoning to hinder its further development. It is still a challenge to seek suitable structure of CoO based catalyst with high activity and stability. The particles could be well dispersed on the carrier by vacuum rotation-evaporation method.²⁸ In this work, CoO nanoparticles are in-situ growing on the g-C₃N₄ to prepare well-dispersed CoO/g-C₃N₄ composite photocatalyst by vacuum rotation-evaporation

and thermal annealing method under nitrogen atmosphere. Corresponding changes have been in the revised manuscript at line 66-74 as follow.

6. There are many previous works about the in-situ synthesis of nanoparticles, such as DOI:10.1016/j.electacta.2018.10.039, 10.1016/j.snb.2019.02.026 and so on. Is there improvement for in-situ synthesis? These need to be explained in Introduction.

Response to Reviewer comment No. 6: The vacuum rotation-evaporation method was used to improve the in-situ synthesis. The particles could be well dispersed on the carrier by vacuum rotation-evaporation method. In this work, CoO nanoparticles are in-situ growing on the g-C₃N₄ to prepare well-dispersed CoO/g-C₃N₄ composite photocatalyst by vacuum rotation-evaporation and thermal annealing method under nitrogen atmosphere. Corresponding changes have been in the revised manuscript at line 70-74 as follow.

Finally, thanks very much for your kind work and precious comments of our paper. On behalf of my co-authors, we would like to express our great appreciation to editor and reviewers.